# Experimental Study of the Microhardness and Microstructure of a Copper Specimen Using the Taylor Impact Test

**Sergey A. Zelepugin *** **, Nadezhda V. Pakhnutova, Olga A. Shkoda and Evgenii N. Boyangin** 

Tomsk Scientific Center of the Siberian Branch of the Russian Academy of Sciences, Tomsk 634055, Russia
* Correspondence: szel@yandex.ru

**Abstract:** One commonly used method for characterizing the dynamic characteristics of materials is the Taylor impact test. This method measures the dynamic yield strength of cylindrical specimens and determines material model constants required for the numerical simulation of the behavior of materials subjected to high-velocity deformation. The purpose of this work is to investigate the microhardness and microstructure of copper specimens at different impact velocities using the Taylor impact test. This paper describes experiments performed on copper specimens (OFHC 99.9%, M1) using a single-stage light-gas gun with impact velocities in the range of 150–450 m/s. After impact, the specimens were cut along the symmetry axis to measure the microhardness and the grain size of the microstructure. Microhardness in the entire area exceeded the initial value for all investigated velocities. The averaged microhardness curves were obtained for each specimen to identify four deformation zones and determine their dimensions depending on the impact velocity. The average grain size in the entire deformed specimen became smaller than in the starting specimen. The study of the microstructure of the specimens has shown that the grain size distribution corresponds to the four deformation zones in the copper specimens.

**Keywords:** Taylor impact test; microhardness; microstructure

## 1. Introduction

In 1946, G.I. Taylor presented methods for characterizing the stress–strain response of materials at high strain rates. A cylinder was impacted onto a massive, rigid anvil in the form of a ballistic pendulum [1]. Taylor found plastic strain in the frontal part of the cylinder, which increased the cross-sectional area and decreased the overall length. The stationary anvil experiment became known as the Taylor impact test (Figure 1) [2–4]. Since then, the Taylor impact test has become one of the primary methods for estimating the dynamic characteristics of materials. This method occupies an intermediate position between the Kolsky method (Split Hopkinson Pressure Bar) [5–7] and high-velocity impact [8–11], and it can estimate the dynamic characteristics of materials in the strain-rate range of $10^4$–$10^5$ s$^{-1}$.

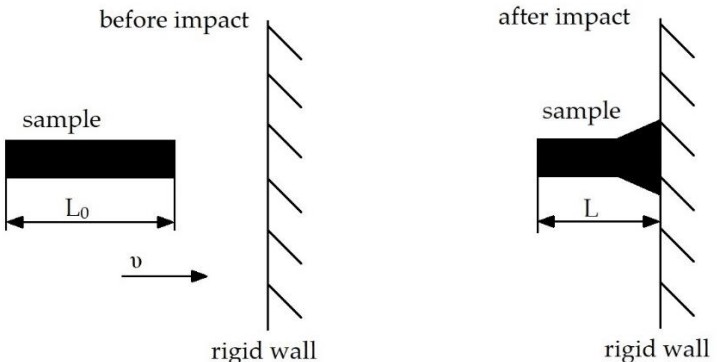

**Figure 1.** Schematic of the classic Taylor impact test.

The "symmetric impact" technique was developed based on the classical Taylor impact test [12–14]. A fixed rod made of the same material and geometry as the impacted rod is used as a rigid support. Impacted and fixed rods deform symmetrically. To be equivalent to the classical Taylor impact test, the rod must be impacted at twice the velocity. The key advantage of the "symmetric" test is that it eliminates the uncertain boundary conditions associated with (a) the displacement of the impact boundary (no anvil is absolutely rigid) and (b) the unknown friction between the rod and the anvil. The "symmetric" test, however, is more difficult to perform than the classical test. The rods must be aligned with such accuracy that the ends are parallel (within a few milliradians) and coaxial (within tens of micrometers).

The Taylor impact test was originally designed to calculate the dynamic yield strength of a cylindrical specimen, taking into account the residual length of the specimen after impact onto a non-deformable target (a rigid wall). This approach has often been used to determine the dynamic yield strength of materials [15–19], as well as to select constitutive relations and constants for numerical simulations [20–29]. At present, experimental studies and numerical simulations of the Taylor impact test have been actively carried out for a wide range of materials and loading conditions. Rodionov et al. [30] applied a combined experimental and numerical approach to study the dynamic plasticity of oxygen-free hard copper (OFHC) for strain rates up to $1.7 \times 10^4$ s$^{-1}$. Lee et al. [31] investigated the strain-hardening of austenitic stainless steel (AISI 304) in a wide range of strain rates from quasi-static to $10^6$ s$^{-1}$. The constitutive relations of metals, including $\alpha$-titanium, copper, $\alpha$-iron, and tantalum materials, are presented in [32] for a wide range of strain rates. Most attention is focused on higher velocities in the Taylor impact tests (solid cylinder) as well as on the simulation of the corresponding deformation characteristics. Gao et al. [33] have developed a modified Taylor impact test using a split Hopkinson pressure bar and high-speed imaging to obtain stress–strain curves. The validity of the proposed method was investigated using experiments and finite element simulation at different impact velocities. A single-shear specimen was used in [34] to investigate the thermo-viscoplastic behavior of aluminum alloy (AA2024-T351) subjected to simple shear stress. Relationships between shear stress and shear strain were obtained over a wide range of strain rates at various temperatures. A combined material model was developed and verified by numerical simulation of the Taylor impact test. Taylor impact tests for projectiles with four types of nose shapes (blunt, hemispherical, truncated ogive, and truncated conical) were performed in [35], and the characteristics of loading conditions were studied depending on the nose shape and impact velocity. Li et al. [36] used the results of [35] to numerically and experimentally study the high-velocity loading of a missile-borne recorder at different velocities using the Taylor impact test. Five plasticity models were considered in [37]: the Johnson–Cook model, the Zerilli–Armstrong model, the Steinberg–Cochran–Guinan–Lund model, the Mechanical Threshold Stress model, and the Preston–Tonks–Wallace model. The Taylor impact test was shown to be sufficient to determine the parameters of the models with acceptable accuracy.

The purpose of this work is to experimentally investigate the microhardness and microstructure of copper specimens at different impact velocities using the Taylor impact test and to apply the obtained data in numerical simulation of high-velocity deformations in the strain-rate range of $10^4$–$10^5$ s$^{-1}$.

## 2. Materials and Methods

Experiments were performed with copper cylindrical specimens (99.9% OFHC, M1) with a length of 34.5 mm, a diameter of 7.8 mm, and a weight of about 15 g. The composition of the specimens is shown in Table 1.

A single-stage light-gas gun (Research Institute of Applied Mathematics and Mechanics, Tomsk State University, Tomsk, Russia) [38], which accelerates specimens by compressed gas (helium) supplied from a gas cylinder, was used in the Taylor impact test experiments presented in this paper. The ballistic stand was used to select impact conditions that provide a specimen velocity in the range of 150–450 m/s. After the experiment,

the specimens were cut into two equal parts along the symmetry axis by a DK7732 CNC EDM wire-cutting machine (P&G Industrial, Guilin, China).

**Table 1.** Composition of specimens.

| Cu | Fe | Ni | Zn | Pb | O |
|------|--------|--------|--------|--------|-------|
| >99.9 | <0.005 | <0.002 | <0.004 | <0.005 | <0.05 |
| **S** | **As** | **Sb** | **Sn** | **Bi** | |
| <0.004 | <0.002 | <0.002 | <0.002 | <0.001 | |

The microhardness of the specimens was measured along an axial line with a PTM-3M microhardness meter (LOMO Co., Saint Petersburg, Russia) using diamond indenters according to GOST 9650-76. The measurement error of this device was 2%. Microhardness values were also calculated according to GOST 9650-76.

The microstructure of the chemically etched specimens was studied with a Zeiss Axiovert 200M fluorescence microscope (Carl Zeiss Industrielle Messtechnik GmbH, Oberkochen Germany). Grain sizes were analyzed by optical microscopy (the secant method) [39]. Based on the results of the measurements, diagrams of the grain size distribution were plotted, and the average grain size and the range of the random variable were calculated.

## 3. Results and Discussion

### 3.1. Microhardness

Figure 2 shows the cross-sections of the copper specimens after impact for different initial velocities. When the diameter of the specimens is equal to or close to the initial specimen, the deformation is close to elastic, including for a velocity of 416 m/s. The transition of the elastic to the plastic zone is accompanied by deformation in the radial direction and, consequently, by an increase in the diameter. Closer to the contact surface, there is a zone of intense plastic deformation, which leads to an area of cylinder fracture during high-velocity impact. There is also a slight asymmetry in the deformation of the specimens due to the peculiarities of the single-stage light-gas gun [38] associated with the non-uniform deformation of polymer rings during the movement of a cylindrical specimen inside the barrel of the light-gas gun, which may cause some difficulties for direct comparison of experimental results with numerical simulation.

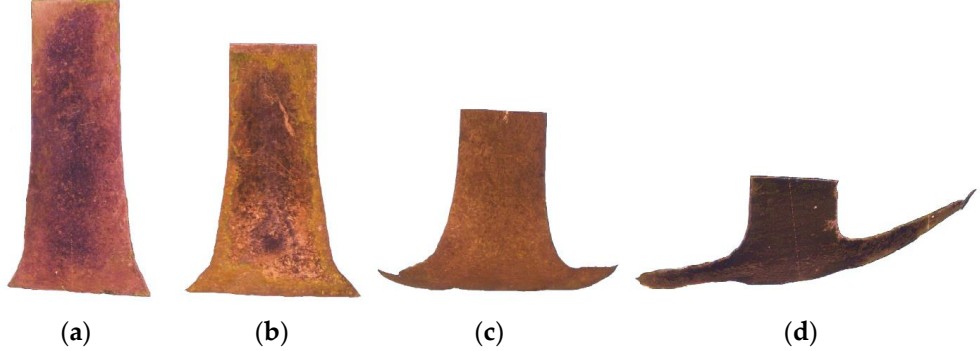

(a)          (b)          (c)          (d)

**Figure 2.** Cross-sections of the copper specimens after impact onto a rigid wall with different initial velocities: (**a**) 162 m/s, (**b**) 225 m/s, (**c**) 316 m/s, and (**d**) 416 m/s.

The measured averaged microhardness of the starting copper specimen is 1150 ± 100 MPa. Such a scatter of values, apparently, is caused by the presence of inhomogeneities in the structure. Figure 3a shows the longitudinal microhardness of the specimen for different impact velocities, and Figure 3b shows the averaged values.

A comparison of the calculated averaged microhardness with the microhardness of the non-deformed specimen (1150 MPa) shows that the microhardness in the entire area exceeds the initial value. Several deformation zones can be distinguished. This is the rear

part of the specimen with a slight increase in the microhardness up to 1250–1350 MPa, apparently due to gas pressure during impact. In the middle part of the specimen, the microhardness reaches the values of a non-deformed specimen. Closer to the contact boundary, the microhardness begins to increase up to 1400–1600 MPa, reaches the area of inflection, and sharply increases up to 1800–2700 MPa. The averaged microhardness is 1172 ± 29 MPa in the rear part of the specimen, 1443 ± 41 MPa in the middle part of the specimen, and 2039 ± 192 MPa closer to the contact boundary of the specimen for a velocity of 316 m/s.

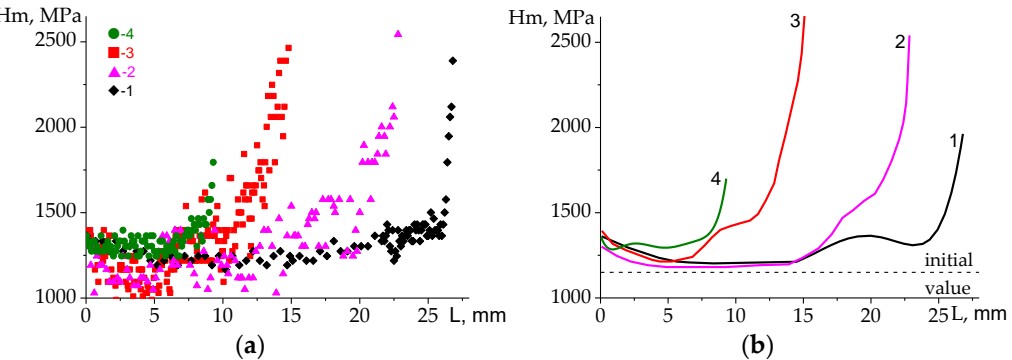

**Figure 3.** Microhardness (**a**) and calculated averaged microhardness (**b**) of the specimens for different impact velocities: (1) 162 m/s, (2) 225 m/s, (3) 316 m/s, and (4) 416 m/s.

The maximum microhardness is observed at an impact velocity of 316 m/s. At higher velocities, the maximum microhardness sharply decreases. The decrease in the microhardness of the specimen at the impact velocity of 416 m/s is caused by its fracture due to the impact and the breakage of the cylinder into fragments.

Consider a specimen with an impact velocity of 316 m/s in detail. For this specimen, two series of measurements were performed along two lines, which are conventionally named C and C1. The location of these lines was chosen based on the following considerations. Line C was placed along the axis of symmetry, and line C1 passed through the middle of the specimen cross-section radius. More than 100 microhardness measurements were performed along each line. This array of microhardness values was averaged, and the results are shown in Figure 4. Figure 4 shows that both curves are non-linear, and the microhardness distributions in both cases are qualitatively and quantitatively similar. Such data can be used to both identify the deformation zones of the specimens and determine their lengths.

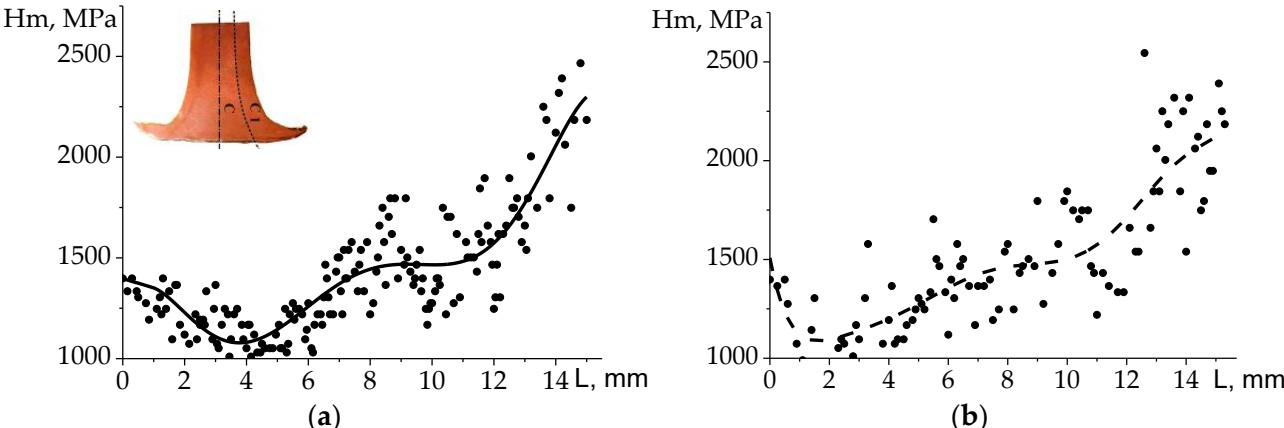

**Figure 4.** Averaged microhardness distribution of the copper specimen along line C (**a**) and C1 (**b**) at the impact velocity of 316 m/s.

### 3.2. Deformation Zones of Cylindrical Specimens

Figure 5 shows four deformation zones in the cylindrical specimen. The length of the deformation zones was determined based on the microhardness distribution. Zone 1 corresponds to near-elastic deformation, Zone 2 corresponds to plastic deformation, Zone 3 corresponds to intense plastic deformation, and Zone 4 corresponds to the fracture of the material. Table 2 shows the lengths of these zones in the specimens depending on the impact velocity.

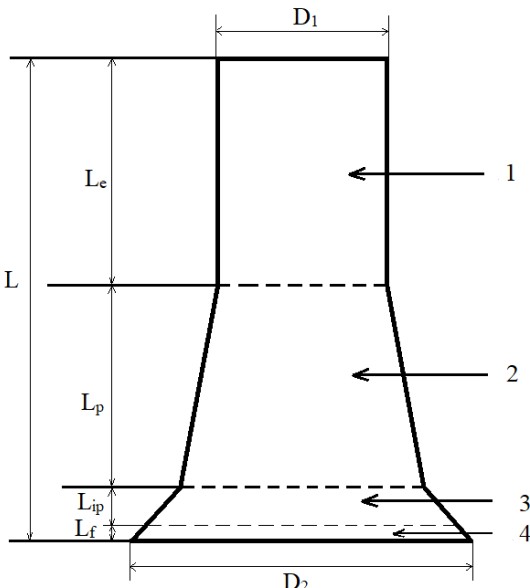

**Figure 5.** Schematic of the specimen deformation after impact onto a rigid wall.

**Table 2.** Lengths of the deformation zones in specimens.

| $\upsilon$, m/s | L, mm | $L_e$, mm | $L_p$, mm | $L_{ip}$, mm | $L_f$, mm | $D_1$, mm | $D_2$, mm |
|---|---|---|---|---|---|---|---|
| 0 | 34.5 | 0 | 0 | 0 | 0 | 7.8 | 7.8 |
| 162 | 26.1 | 12.34 | 10.63 | 3.13 | 0 | 7.8 | 12.7 |
| 225 | 22.5 | 9.78 | 9.51 | 2.28 | 0.93 | 7.8 | 15.8 |
| 316 | 16.1 | 5.87 | 6.83 | 2.26 | 1.14 | 7.9 | 21.4 |
| 416 | 9.3 | 3.5 | 1.38 | 2.41 | 2.01 | 7.9 | 31.28 |

Here, $\upsilon$ is the impact velocity, L is the final length of the specimen after impact, $L_e$ is the length of the elastic deformation zone, $L_p$ is the length of the plastic deformation zone, $L_{ip}$ is the length of the intense plastic zone, $L_f$ is the length of the fracture zone, $D_1$ is the diameter of the rear part of the cylinder, and $D_2$ is the diameter of the contact boundary.

The data presented in Table 2 show that the specimen is not broken at an impact velocity of 162 m/s. This specimen is an example of the classic Taylor impact test, which can be used to develop an adequate numerical model of the impact of a cylindrical specimen onto a rigid target and to select material model constants. All four zones are observed in the specimens at 225 and 316 m/s, but the fracture zone is small compared to the specimen with an impact velocity of 416 m/s. A small plastic deformation zone $L_p$ is observed in the cylinder after the test at an impact velocity of 416 m/s. At this impact velocity, the elastic deformation zone $L_e$ quickly transforms into an intense plastic deformation zone $L_{ip}$ combined with the fracture zone $L_f$.

Longitudinal strain versus impact velocity is shown in Figure 6a. Longitudinal strain is given by $\Delta L/L_0 = (L_0 - L)/L_0$, $L_0$ is the initial length, and L is the length after impact. The velocity value of 416 m/s is not taken into account, as in this case the cylinder is destroyed and the calculation technique cannot be applied. It can be seen that the dependence has a linear character. Figure 6b shows the $L/L_0$ strain graph, which also has a linear character.

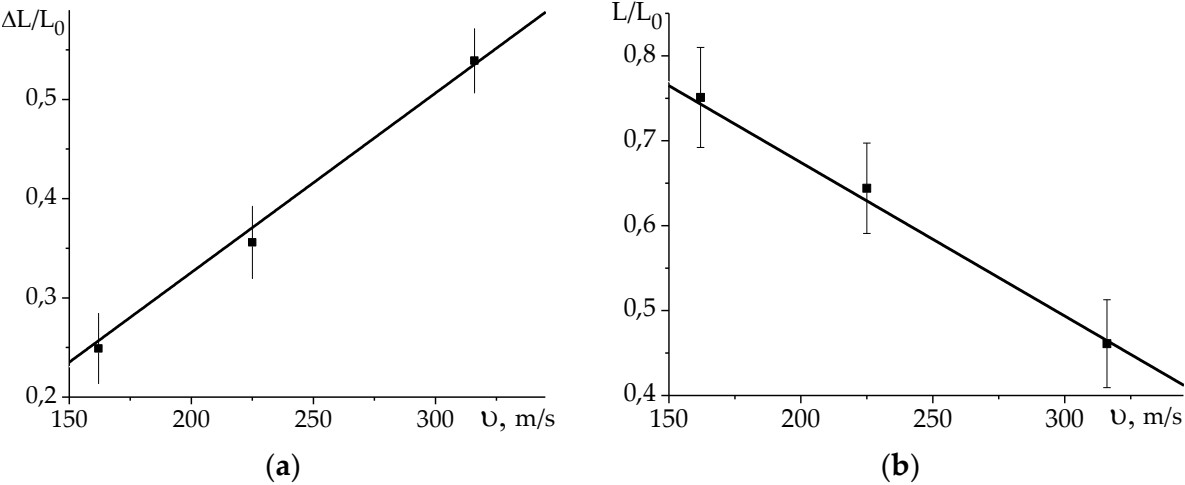

(**a**)  (**b**)

**Figure 6.** Longitudinal strain (**a**) and the ratio of the final length to the original length (**b**) versus the impact velocity of the specimen.

*3.3. Microstructure*

A specimen with all four deformation zones at the impact velocity of 316 m/s was selected to study the deformed microstructure. Figure 7 shows the microstructure of the starting specimen. The grains are elongated along the rolling direction. The rolling direction is indicated by an arrow in Figure 7. The average grain size in the non-deformed specimen was 207.6 μm.

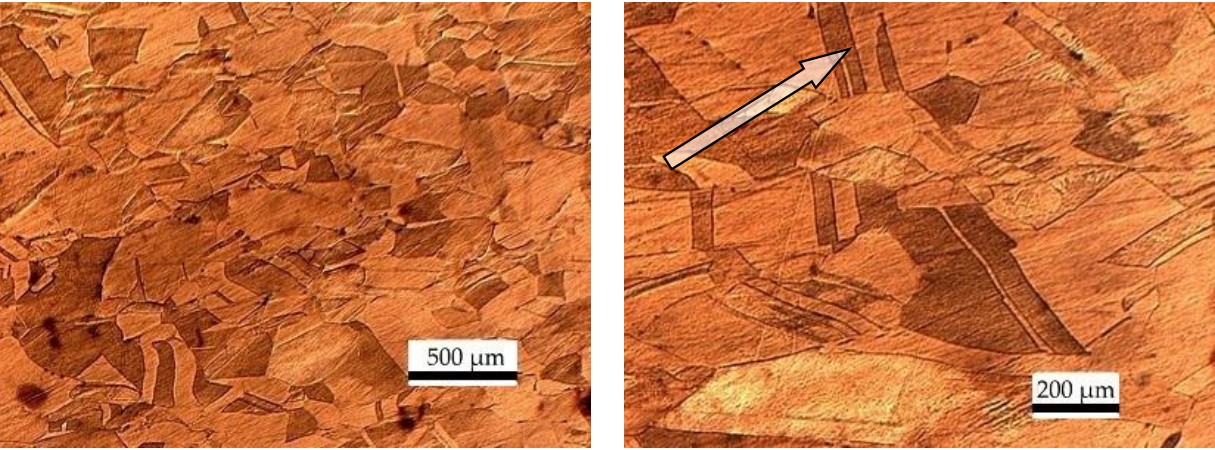

**Figure 7.** Microstructure of a non-deformed specimen.

Figure 8 shows the microstructure in the elastic deformation zone. This image shows a structure almost identical to that of the starting specimen, but the average grain size in this area has decreased to 177 μm. Figure 9 shows a histogram of the grain size distribution in the elastic deformation zone. The grain distribution shown in the histogram is homogeneous with a shift to the fine-grained region. Grains 30–140 μm in size predominate, but the average grain size is 177 μm.

The number of deformation bands increases and the potential of twin formation grows in the zone of plastic deformation (Figure 10a), which is a typical phenomenon during the deformation of copper. Formation of deformation bands was caused by sliding dislocation lines due to exceeding the yield strength and transition of the material into the plastic deformation zone. Grain size decreases to 122.3 μm. Wave-like lines appear near the region of intense plastic deformation inside the grains. This is a consequence of the dislocation of slip lines during compression and shear deformation. Figure 10b demonstrates the

microstructure in the zone of severe plastic deformation, in which localized shear bands form and rotational deformation modes are observed. The average grain size in this area decreases to 75.7 µm. In the fracture zone, the average grain size decreases to 37 µm in the pre-contact zone and to 17.8 µm in the contact zone.

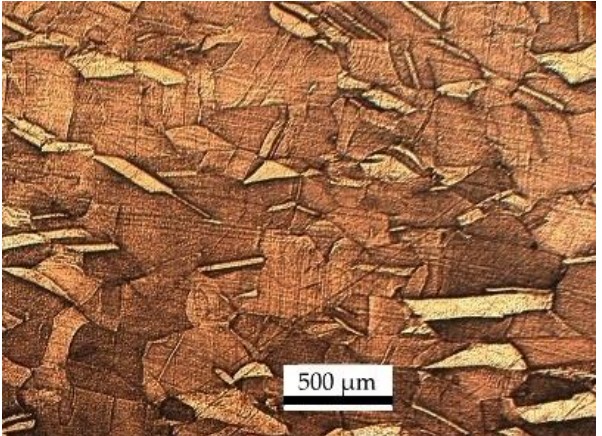 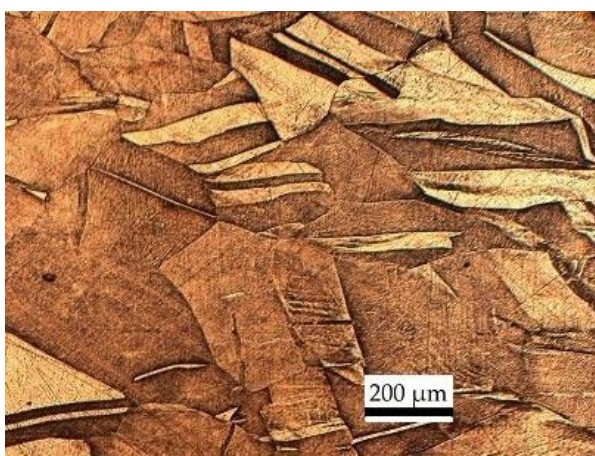

**Figure 8.** Microstructure of the elastic deformation zone for an impact velocity of 316 m/s.

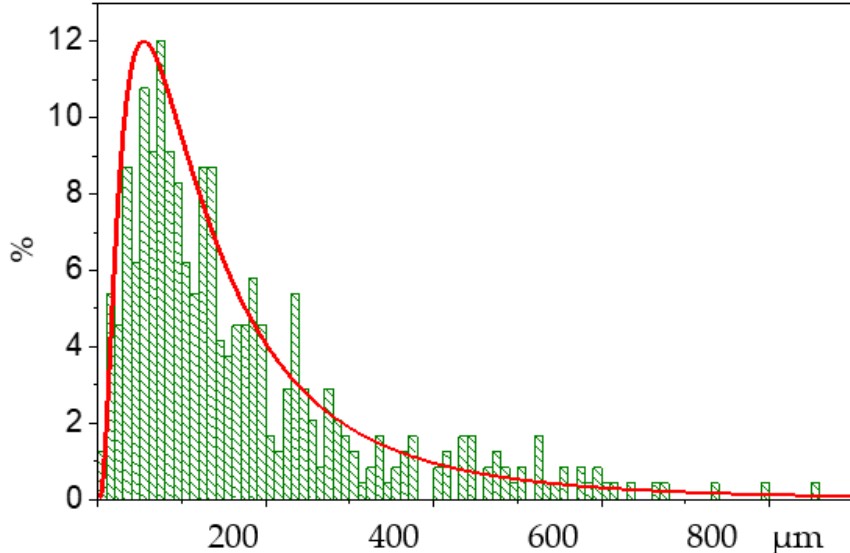

**Figure 9.** Diagram of grain size distribution in the zone of elastic deformation.

Figure 10c shows the microstructure of the contact zone between the specimen and the rigid wall. The structure changes in the contact zone, and a wave structure is formed, which differs sharply from the structure in other zones. This phenomenon is explained by the extreme plastic deformation of material due to the shockwave and deformation processes. Longitudinal shock-wave loading causes deformation both in axial and radial directions, which is accompanied by heating, shear extrusion of material, and further cooling and recrystallization.

Figure 11 summarizes the measurements of the average grain size in the non-deformed specimen and in the different zones of the deformed specimen. It should be noted that the average grain size in the elastic deformation zone is smaller than in the non-deformed specimen. The average grain size monotonically decreases towards the impact zone. In zone 4 (fracture of the specimen), the grain structure is subdivided into two subzones: pre-fracture zone (pre-contact zone) 4a and fracture zone (contact zone) 4b, where the grain structure transforms into a wave structure.

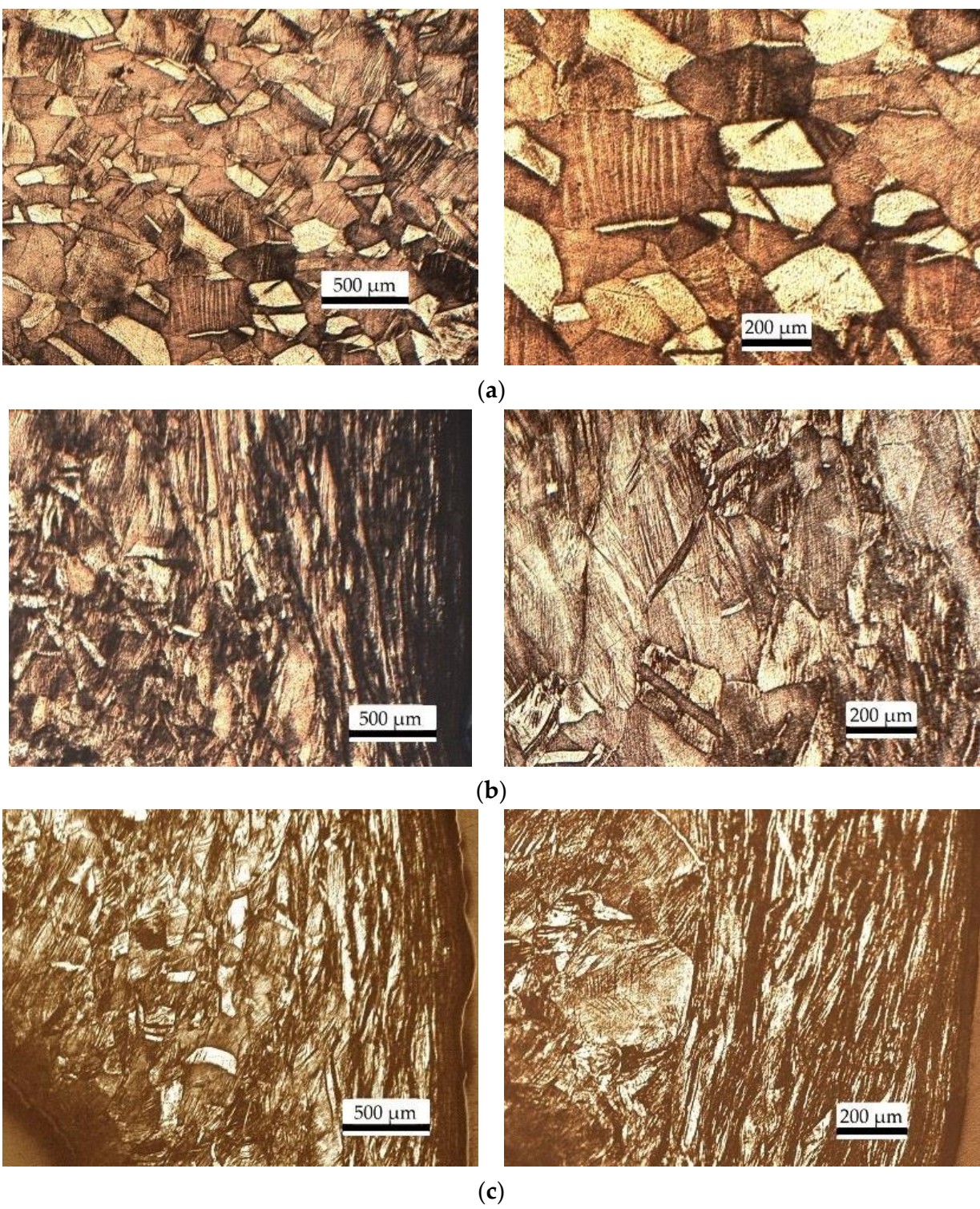

**Figure 10.** Microstructure in the plastic deformation zone (**a**), intense plastic deformation zone (**b**), and fracture zone (**c**) of the specimen for an impact velocity of 316 m/s.

The studies have shown that the passage of a shock wave through a copper cylinder leads to structural transformations of the material, changing its physical and mechanical properties. After the impact of the cylinder onto a rigid wall, the average value of microhardness in the whole specimen exceeds the initial value equal to 1150 MPa, and in the zone of impact, a considerable increase in microhardness up to 1800–2700 MPa is observed.

This indicates deformation hardening not only in the head of the copper specimen but also in its entire volume.

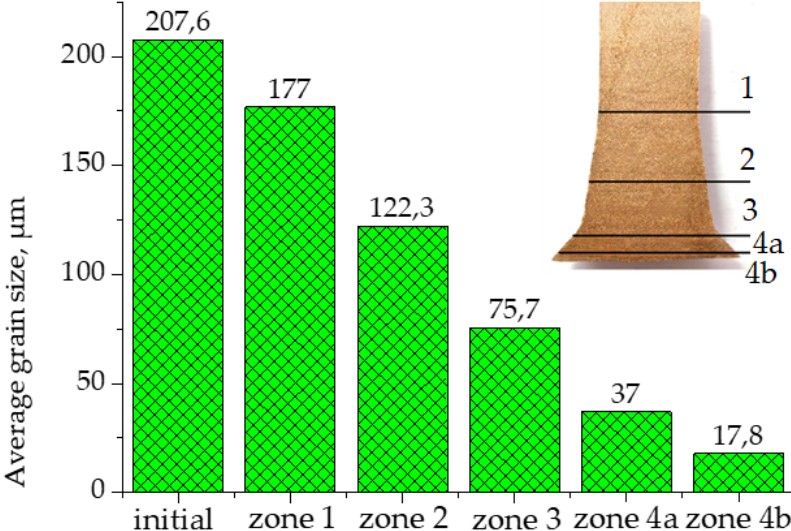

**Figure 11.** Average grain size in the non-deformed specimen and in the different zones of the deformed specimen for an impact velocity of 316 m/s.

The microhardness analysis of the deformed cylinder has revealed four deformation zones: elastic deformation, plastic deformation, intense plastic deformation, and fracture. The lengths of these zones were determined for different impact velocities. No fracture zone was found at the velocity of 162 m/s. All four deformation zones were clearly visible at impact velocities of 225 and 316 m/s. At a velocity of 416 m/s, the zone of moderate plastic deformation was practically absent, and the forepart of the specimen was broken into fragments. The dependence of the longitudinal strain on the impact velocity is linear for a velocity range of up to 316 m/s, and the forepart of the specimen is broken into fragments at 416 m/s, which does not allow this case to be taken into account for the calculations. The longitudinal strain of the cylinder is 0.53 at a velocity of 316 m/s, 0.35 at a velocity of 225 m/s, and 0.24 at a velocity of 162 m/s.

The microstructural analysis of the specimens after impact onto a rigid target also showed four zones, which can be distinguished by the intensity of elastic–plastic deformation of the material. The average grain size in the tail part of the specimen is smaller than in the non-deformed specimen. This change can be explained by the passage of a shock wave, which is undetectable in optical images of the microstructure and can be detected only by statistical methods. The average grain size monotonically decreases towards the impact zone. Two subzones appear in the contact area. One subzone contains material with a granular structure and another subzone, at the contact boundary, has a structure that becomes wavy, without the ability to distinguish individual grains.

The microhardness of the deformed copper specimens in the impact velocity range of 162–416 m/s and the microstructure of the specimen for a velocity of 316 m/s show strain hardening and interrelated refinement of the grain structure throughout the specimen. This also applies to the tail part of the cylinder (Figure 5, zone 1), which can be called "elastically deformed" only approximately.

The data presented in the paper can be used to assess the adequacy of the physical and mathematical model for numerical simulation of the high-velocity deformation of metals and alloys.

## 4. Conclusions

This paper presents experimental data for cylindrical copper specimens (99.9% OFHC) impacted onto a rigid wall (Taylor impact test). The configurations of the specimens

after impact with velocities in the range of 162–416 m/s are presented. The distribution of microhardness in the axial cross-section of the specimens has been obtained. The distribution of deformation zones in the specimens has been shown. The microstructure of the specimens has been examined, and the average grain size has been determined. The conclusions are as follows:

1.  Analysis of the microhardness distribution and the average grain size of the microstructure has revealed four deformation zones of copper specimens after impact onto a rigid wall: a zone of deformations close to elastic, a zone of plastic deformations, a zone of intense plastic deformations, and a zone of material fracture. The lengths of the deformation zones have been determined depending on the impact velocity;

2.  Microhardness and microstructure have been found to vary throughout a cylindrical copper specimen in the impact velocity range of 162–416 m/s. This also applies to the first zone of deformation, the region of approximately elastic deformation, where the microhardness value is higher than the initial value by an average of 9–12%. The average grain size is lower by 15% at an impact velocity of 316 m/s;

3.  The maximum average value of microhardness in the copper specimen after impact with a rigid target reaches 1800–2700 MPa at an impact velocity of 316 m/s, which is 1.6–2.3 times higher than the initial value. The minimum average grain size of the specimen microstructure at the impact velocity of 316 m/s reaches 17.8 μm, which is 11.7 times smaller than the initial value.

**Author Contributions:** Conceptualization, S.A.Z. and N.V.P.; methodology, S.A.Z., E.N.B. and O.A.S.; validation, S.A.Z.; formal analysis, N.V.P., E.N.B. and O.A.S.; investigation, N.V.P., E.N.B. and O.A.S.; resources, S.A.Z.; data curation, S.A.Z.; writing—original draft preparation, N.V.P.; writing—review and editing, S.A.Z. and O.A.S.; visualization, N.V.P., E.N.B. and O.A.S.; supervision, S.A.Z.; project administration, S.A.Z.; funding acquisition, S.A.Z. All authors have read and agreed to the published version of the manuscript.

**Funding:** The work was supported by the Ministry of Science and Higher Education of the Russian Federation (project No. 121031800149-2).

**Data Availability Statement:** Not applicable.

**Acknowledgments:** The authors are grateful to Yu.F. Khristenko, (NIIPMM TSU, Tomsk) for help in conducting experiments using ballistic test facility.

**Conflicts of Interest:** The authors declare no conflict of interest.

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
