# Peer review of "Experimental Study of the Microhardness and Microstructure of a Copper Specimen Using the Taylor Impact Test"

_metals, doi:10.3390/met12122186_

Round 1

Reviewer 1 Report

I congratulate the authors on the interesting results of analyzes at the microstructure level of the Taylor impact tests. I suggest using these results in the continuation of research related to constitutive modeling on a physical basis.

Author Response

Thank you very much for your high appreciation. Your comments are very helpful for our future work.

Author Response

The authors are very grateful to the reviewer for this detailed and informative review. We have carefully checked all comments and have provided a response to each item for your consideration. The changes are highlighted in color in the corrected manuscript. We hope that these improvements are satisfactory and that the manuscript now meets the standards of the journal Metals.

Reviewer 3 Report

1. It is recommended to revise the manuscript for typo errors and grammar mistakes.

2. Figure 1 is original? Or adopted from any published article? If it not original, pleasesubmit the copyright for the same.

3. It is recommended to use author name et al. instead of using In [31], the same is repeated in many places. Please revise them as well.

4. Figure 2 caption is misleading, remove the – sign in the numbers, it feels like a negative value.

5. It is recommended to represent the microhardness along with the average error bars in microhardness.

Author Response

(The authors gave the same response as above.)

Reviewer 4 Report

In this paper, the microhardness and microstructure of copper at different impact velocities using the Taylor impact test are systematically studied, and some valuable results are provided. However, the following problems need to be solved before publication:

1.      The reason why the microhardness and microstructure of copper should be studied by the Taylor impact test method is not clearly stated in the introduction section.

2.      “in Ref.” appeared on Page 2 many times, language polishing is suggested.

3.      All tables should be in the format of a three-line table.

4.      The syntax error is shown in lines 115, 162, 187 and 189. In addition, some sentences in the paper are too long and also have grammatical errors, such as the sentence in lines 164-170. They should be written separately.

5.      The "Peculiarities of impact" in line 117 and the "structural features of a material" in line 125 should specify the characteristics.

6.      The caption in lines 128, 153, 216 and 192 should be written before the description about the phenomenon of the figure.

7.     Page 4, line 143. “For example, an impact velocity of 316 m/s; at higher velocities, there is a sharp decrease...”. ";" is used to connect two sentences in many places. Generally speaking, the connectives (and) are selected to connect the two sentences.

8.     Some paragraphs (only two sentences) in the text can be merged with related paragraphs. For example, lines 158-160 can be merged with lines 147-153.

9.     Page 6, line 180. “For example, has almost no plastic deformation zone; at this impact velocity, the elastic deformation zone quickly becomes a zone of intense plastic deformation...”. This sentence is difficult to understand. The sentence mentions that there is no plastic deformation, but also say that the elastic deformation quickly becomes plastic deformation.

10.  The rolling direction (line 197) should be marked in the Figure 7, and different orientations cannot be seen in the Figure 7.

11.   The ruler positions of Figures 7, 8, 10, 11, and 12 need to be unified. In my opinion, a paragraph in the article should be rich in content, thus it is necessary to combine Figures 8, 10, 11 and 12 into a single figure and the relevant paragraphs should also be combined. With regard to Figure 9 it is usually necessary to add a description of whether the grain distribution is homogeneous (line 210).

12.   The data for grain size in lines 198-220 are obtained from Figure 13, thus it is necessary to write the description of Figure 13 (lines 241-247) before line 196. When describing something similar, such as grain size at different impact velocities (lines 213, 218 and 219), we need to change the syntax to add the richness of language.

13.   Since the color of Figure 10 is black and white, it is not easy to see the deformation bands and twins in the figure (line 212). Thus, it is necessary to mark them in the Figure 10. Furthermore, the reasons for their formation need to be explained.

14.   Some explanations on the causes of phenomena need to be supported by references (lines 234-237). “For example, this is a consequence of the dislocation slip lines…”.

15.  Conclusion: please make a summary in brief (selecting the most important points to write) and then elaborate each conclusion (not too long). In addition, the contents in the second and third conclusion are repeated.

16.   The author is requested to carefully revise the language of the full text. Some incomprehensible sentences appear in the text.

Author Response

(The authors gave the same response as above.)

Round 2

Reviewer 4 Report

The manuscript can be accepted in present form.